# Evaluation of the Approach towards Vaccination against COVID-19 among the Polish Population—In Relation to Sociodemographic Factors and Physical and Mental Health

**DOI:** 10.3390/vaccines11030700

**Published:** 2023-03-19

**Authors:** Justyna Gołębiowska, Anna Zimny-Zając, Mateusz Dróżdż, Sebastian Makuch, Krzysztof Dudek, Grzegorz Mazur, Siddarth Agrawal

**Affiliations:** 1Department and Clinic of Internal Medicine, Occupational Diseases, Hypertension and Clinical Oncology, Wroclaw Medical University, 50-556 Wroclaw, Poland; ju.golebiowska@gmail.com (J.G.);; 2Medonet, Ringier Axel Springer Poland, Domaniewska St. 49, 02-672 Warsaw, Poland; 3Faculty of Medicine, Wroclaw Medical University, J. Mikulicza-Radeckiego 5, 50-345 Wroclaw, Poland; 4Department of Clinical and Experimental Pathology, Wroclaw Medical University, 50-367 Wroclaw, Poland; 5Statistical Analysis Center, Wroclaw Medical University, 50-368 Wroclaw, Poland

**Keywords:** attitude toward COVID-19 vaccination, sociodemographic factors, COVID-19 vaccination

## Abstract

Due to the rapid development of COVID-19 vaccines, the world has faced a huge challenge with their general acceptance, including Poland. For this reason, we attempted to determine the sociodemographic factors influencing the decision of positive or negative attitudes toward COVID-19 vaccination. The analysis included 200,000 Polish participants—80,831 women (40.4%) and 119,169 men (59.6%). The results revealed that the most common reasons for vaccine refusal and hesitancy were the fear of post-vaccination complications and their safety (11,913/31,338, 38.0%; 9966/31,338, 31.8%). Negative attitudes were observed more often among male respondents with primary or secondary education (OR = 2.01, CI95% [1.86–2.17] and OR = 1.52, CI95% [1.41–1.63], respectively). On the other hand, older age ≥ 65 (OR = 3.69; 95%CI [3.44–3.96]), higher education level (OR = 2.14; 95%CI [2.07–2.22]), living in big cities with a range of 200,000–499,999 inhabitants and more than 500,000 inhabitants (OR = 1.57, CI95% [1.50–1.64] and OR = 1.90, CI95% [1.83–1.98], respectively), good physical conditions (OR = 2.05; CI95% [1.82–2.31]), and at last normal mental health conditions (OR = 1.67, CI95% [1.51–1.85]) were significantly associated with COVID-19 vaccine acceptance. Our study indicates which population group should be further supplied with data and information by health education, the government, and healthcare professionals to alleviate the negative attitude toward COVID-19 vaccines.

## 1. Introduction

The Coronavirus disease 2019 (COVID-19) caused by the SARS-CoV-2 virus has led to a dramatic public health problem since its first outbreak at the beginning of 2020. Confirmed cases of this disease reached approximately 700 million by February 2023, with a death toll exceeding more than 6 million worldwide. In Poland, the total number of confirmed cases reached more than 6 million, with over 120,000 deaths [1]. Undoubtedly, this alarming increase in the number of COVID-19-associated morbidities and mortalities had an unprecedented negative impact on economic activity, education, travel, international trade and transport, global production, distribution, social activities, and healthcare [2,3]. Health services and researchers worldwide were working under severe pressure to provide the public best available care. Other comorbidities (e.g., cancer [4], chronic kidney disease [5], heart disease [6], and diabetes mellitus [7]) and sociodemographic factors (e.g., older age [8], stress [9], and obesity [10]) have the greatest impact on the risk of severe COVID-19 complications. 

To date, despite the unprecedented efforts of scientists, there is no successful treatment strategy for COVID-19 infection. Therefore, the urgent COVID-19 vaccine development and vaccination campaigns are the only breakthrough in the fight against the SARS-CoV-2 virus and the primary exit strategy from this global crisis. In November 2020, approximately one year after the COVID-19 outbreak, at least 55 vaccines were undergoing clinical trials on humans, and at least three were approved for public use. As estimated by Anderson et al., at least 60–72% of the population has to be vaccinated to herd immunity, which could significantly lead to SARS-CoV-2 eradication [11]. However, the approach to outrun the virus mutation, immunize the majority of the population, and stop its spread has led to unexpected results. Due to the accelerated vaccine development, there has been a loss of confidence in their safety and effectiveness. According to the World Health Organization (WHO), the hesitancy to take a COVID-19 vaccination is among the top ten global threats [12]. It is, however, worth keeping in mind that the willingness to vaccinate varies in different countries and can be shaped by various factors (e.g., history of diseases, sociodemographic factors, and society). For instance, Lazarus et al., surveyed 13,426 randomly selected individuals from 19 countries to determine the willingness to accept a COVID-19 vaccination. The highest acceptance rate (more than 80%) was observed in Asian nations (including China, the Republic of Korea, and Singapore) and middle-income countries, such as Brazil, India, and South Africa. In contrast, the lowest willingness to take the COVID-19 vaccination was observed in Poland and Russia (around 55%) [13]. Interestingly, according to the Public Opinion Research Center in Poland, there was an increase in the intention to be vaccinated against COVID-19 over time. Compared to November 2020, the percentage of these individuals increased by nearly 67% [14]. Taking into account only European countries, Portugal is at the top in the ranking of people taking the COVID-19 vaccination (94.9%%—in 2023) [15]. Therefore, constant monitoring of the statistical data in this regard, especially in countries with relatively low acceptance rates in the global perspective, is highly beneficial to implement proper strategies for COVID-19 vaccination programs and achieve the public healthcare success [15]. 

Poland is one of the countries offering vaccination to its populations in a phased manner. The vaccination program in Poland started on 27 December 2020, when the first vaccine was delivered the Polish healthcare facility [16]. Initially, vaccines were directed to senior citizens at the age of 80 or more [16], as this age group is at the highest risk of COVID-19 complications. Subsequently, other Polish citizens categorized by age had access to free-of-charge vaccination. To date, several studies from Poland brought up the problem of concerns of COVID-19 vaccination. For instance, Sowa et al. determined that the fear of side effects is the reason for refusion of COVID-19 vaccination for 82% of a study group [17]. Similarly, Stasiuk et al. concluded that the main arguments against vaccinations were as follows, among others: (1) no proven effectivity, (2) low quality of the research on vaccines, and (3) the intention of the pharmaceutical industry and medical profession [18]. Furthermore, Walkowiak et al. showed that a low level of education is a negative predictor of COVID-19 vaccine acceptance [19]. Other factors influencing the vaccination acceptability seem to be religion, national narcissism, and conspiracy theories [17]. According to Raciborski et al., Polish females had higher odds of refusing COVID-19 vaccination compared with males, suggesting that age is another factor playing role in this regard [20].

Understanding the concerns and factors triggering a decision not to be COVID-19 vaccinated can be useful to overcome vaccine hesitancy. For this reason, we utilized the National Test for Poles’ Health (NTZP)—an online study performed yearly since 2020, collecting data from a large group of Polish Internet users. The decision to vaccinate may result from culture, beliefs, or sociodemographic characteristics. Based on data collected in the NTZP, the current study aims to determine sociodemographic factors contributing to the attitude to vaccination against COVID-19. We believe our study is one of the stepping stones to target the groups of Polish citizens with the highest risk of vaccine hesitancy. Including other reports from Poland, our study is crucial to be considered while developing strategies to strengthen the COVID-19 vaccination programs and educational interventions. Furthermore, it may be used by healthcare agencies in different countries willing to re-align their vaccination programs and target groups with the most negative attitudes to COVID-19 vaccination. 

## 2. Materials and Methods

### 2.1. Study Design

The National Test for Poles’ Health (NTPH) is a valuable information source on Polish Internet users’ health. Thus far, it has been conducted in three waves (2020, 2021, and 2022) [21]. The questionnaire was filled out in Polish by over 970,000 respondents in all three waves [22]. It was distributed online via a social networking site. The survey was fully anonymous and voluntary. For the purpose of this particular study, responses from one wave (2022) were analyzed—a representative sample of 200,000 adults. The scheme of the online survey is shown in Appendix A; it was translated from Polish to English for reader’s understanding. The evaluated sample of the study group was obtained by stratified sampling per the voivodeship demographic structure of Poland. The duration of the survey ranged from 15 to 20 min. All participants provided informed consent for collecting the data and were informed about the goal of the survey. Participation in the study provided no compensation.

### 2.2. Explanatory Variables

The online survey used in the study included questions regarding the respondent’s sociodemographic data and questions necessary to evaluate the attitude toward COVID-19 vaccination (Appendix A). Sociodemographic data included: (1) gender (male or female), (2) age (categorized as 18–24, 25–34, 35–44, 45–54, 55–65, 65, and more), (3) education (primary, secondary, or higher), (4) place of residence (village; town less than 19,000 inhabitants; town between 20,000 to 49,000 inhabitants; town between 50,000 to 99,000 inhabitants; town between 100,000 to 199,000 inhabitants; town between 200,000 to 499,000 inhabitants; and town more than 500,000 inhabitants), and (5) region of Poland (south, northwest, southwest, north, central, east, and Masovian voivodeship). Furthermore, to determine BMI levels (kg/m^2^), respondents were asked to provide body weight (kg) and body height (cm), allowing us to categorize their weight into (1) underweight, (2) normal, (3) overweight, and (4) obese. Additionally, respondents were asked to rate subjective physical and mental health on a five-point Likert scale, choosing from “excellent” to “very bad” (Table 1). 

### 2.3. Measures

The survey included three questions (Appendix A): (1) Are you vaccinated against influenza?; (2) Will you get vaccinated against COVID-19?; (3) Why do you not want to get vaccinated against COVID-19?; The evaluation of the attitude toward influenza vaccination in the study group was measured by counting points obtained while answering the abovementioned questions (Appendix A, question 1 (Q1)). The fewer points received, the more positive attitude towards vaccination, and vice versa; the more points obtained, the more negative attitude towards vaccination. The measurement of patients who were vaccinated against seasonal influenza was made based on the honest and reliable participants’ responses, which we believe are consistent with the actual truth. In the case of COVID-19 vaccination evaluation (Appendix A, question 2 (O_2_)), the positive attitude was measured by answering “1—I have already been vaccinated” or “2—I intend to take a COVID-19 vaccination”. The negative attitude to COVID-19 vaccination was measured by answering “3—I don’t know yet” or “4—No, never” (Table 2). Additionally, among respondents with a negative attitude toward COVID-19 vaccination, we asked about the reason for that statement (Appendix A, question 3 (Q3)). Respondents could select the argumentation from: “I have concerns about the safety of the COVID-19 vaccine”, “I am afraid of post-vaccination complications”, “I can’t get vaccinated due to medical reasons”, and “I am against vaccination in general”. The dichotomization of the answers to the question about the attitude to COVID-19 vaccination allowed us to estimate the odds ratios and highlight significant predictors of their statements. 

### 2.4. Statistics

Nominal and ordinal variables are presented in the contingency tables as numbers (*n*) and percentages (%). Spearman’s rank correlation coefficient (rho) and Pearson’s chi-square test were used to assess the relation between two ordinal variables. Odds ratios and their 95% confidence intervals were also calculated for the 2 × 2 tables. Significant predictors of negative or positive attitudes towards COVID-19 vaccination were those whose odds ratios were outside the range of 1.5 times the reference values. Statistical software package STATISTICA v. 13.3 (TIBCO Software Inc., Palo Alto, CA, USA) was used for the analysis.

## 3. Results

### 3.1. Study Group

The analysis included 200,000 participants—80,831 women (40.4%) and 119,169 men (59.6%). Of the total respondents, 26.4% were older respondents aged more than 65 years old. The majority of the participants had higher education (56.2%). The place of residence was quite evenly distributed; a similar percentage of the study group lived either in a village or in a large city with more than 500,000 inhabitants (22.1% and 17.4%, respectively). Only 1.7% of respondents were underweight, and more than half of the study population had excessive body weight: 38.8% were overweight, and 25.2% suffered from obesity. Most respondents defined their physical and mental health status as “good” (40.2% and 41.8%). Further characteristics of the subjects included in this study can be found in Table 1.

At the time of the study (2022), most respondents declared COVID-19 vaccination (83.2%); 6.7% were uncertain about taking the vaccine, and 8.9% maintained a negative attitude (Table 2). To clarify if the negative attitude refers to COVID-19 vaccination specifically or if is it a general opinion, respondents were also asked about their willingness to take an influenza vaccination. The vast majority of the study group was never vaccinated against influenza (72.2%; Q1, answer 7 in Table 2, Figure 1). However, 11.3% of respondents declared taking an influenza vaccination regularly (11.3%; Q1, answer 1 in Table 2, Figure 1).

Due to the observed positive relation between the attitude to COVID-19 vaccination and influenza vaccination (rho = 0.219, *p* < 0.001), we were able to determine how these approaches were changing under the influence of each other. Among those vaccinated against COVID-19, the number of respondents regularly taking an influenza vaccination increased from 11.3% (Table 2) to 13.3% (Figure 2). Furthermore, among those denying the COVID-19 vaccination, only 0.7% of them declared taking an influenza vaccination, including in 2022.

### 3.2. Predictors of a Positive Attitude toward COVID-19 Vaccination

According to our study, the positive attitude toward COVID-19 vaccination increases with age. Respondents over 65 years old were nearly four times more likely to declare positive approaches to COVID-19 vaccination than respondents aged 18–24 (OR = 3.69 CI = 95% [3.44–3.96], Table 3). A slightly lower but statistically significant relation was observed among respondents aged 55–64 compared to those aged 18–24 (OR = 2.17, CI = 95% [2.03–2.33], Table 3). Furthermore, respondents with higher education compared to those with primary education were approximately two times more likely to declare positive approaches to COVID-19 vaccination (OR = 2.14, CI = 95% [2.07–2.22], Table 3). In addition, people living in large cities (200,000–499,999 inhabitants and more than 500 000 inhabitants) were more likely to take the vaccine (OR = 1.57, CI95% [1.50–1.64] and OR = 1.90, CI95% [1.83–1.98], respectively, Table 3). Among respondents in good physical condition, there was a twofold increased likelihood to declare a positive attitude toward COVID-19 vaccination compared to those in very bad physical condition (OR = 2.05, CI = 95% [1.82–2.31], Table 3). Additionally, we found a statistically significant relation among respondents in other than good physical condition. Still, it never achieved the twofold change compared to those declaring very bad physical condition (Table 3). The same situation was observed among respondents in very good, good, and normal mental conditions—they were more likely to declare a positive attitude toward COVID-19 vaccination than those in very bad mental condition, but this likelihood was less than twofold (for instance, very good mental condition—OR = 1.62, CI95% [1.46–17.79], good mental condition—OR = 1.89, CI95% [1.71 = 2.09], and normal mental condition—OR = 1.67, CI95% [1.51–1.85], Table 3). We did not observe a significant relation between gender and region in Poland (Table 3). The odds ratios and the 95% confidence intervals for statistically significant predictors of a positive attitude toward COVID-19 vaccination are shown in Figure 3.

Thus, the positive attitude toward COVID-19 vaccination was observed predominantly among older respondents with higher education, living in large cities (at least 200,000 inhabitants), and declaring good physical and mental condition (Table 3). 

### 3.3. Predictors of Negative Attitude toward COVID-19 Vaccination

The rationale for refusing vaccination against COVID-19 (Q3, Table 2) was provided by 15% of the study population (31,338/200,000 respondents). The most frequently cited reasons were fear of post-vaccination complications and concern about their safety (38.0% and 31.8%, respectively). Furthermore, 4372 respondents could not be COVID-19 vaccinated due to medical reasons (16.2%), and 5087 respondents declared to be against vaccinations in general (14.0%, Table 2). 

The likelihood of being against COVID-19 vaccination was more than twofold higher among men than women (*p* < 0.001, OR = 2.20, CI = 95% [2.07–2.34], Table 4). We may assume that education status plays a crucial role in the decision-making process. Respondents with primary or secondary education were more likely to declare anti-vaccine attitudes (*p* < 0.001, OR = 2.01, CI = 95% [1.86–2.17] for primary education, and *p* < 0.001, OR = 1.52, CI = 95% [1.41–1.63] for secondary education, Table 4). Furthermore, respondents declaring very good physical health status were approximately 1.5 times more likely to report anti-vaccination approaches compared to those with very bad status (*p* < 0.001, OR = 1.51, CI95% [1.15–1.99], Table 4). There was no statistically significant relation between negative vaccination attitude, place of residence, region in Poland, and BMI.

Overall, reluctance toward COVID-19 vaccination was observed mainly among men with primary and secondary education declaring very good physical condition (Table 4). 

## 4. Discussion

Our study is one of the largest population-based studies (*n* = 200,000 participants) addressing attitudes toward vaccination in the context of the COVID-19 pandemic in Poland. Furthermore, to the best of our knowledge, it is the most up-to-date study on attitudes toward COVID-19 vaccines in Poland. Collected data show that 83.2% of the respondents were COVID-19 vaccinated. However, the percentage applies only to the adult population of Poland. Official updates from the Polish Ministry of Health show that 60.6% of the total population (67.4%, 18+ year) was vaccinated with at least one dose against COVID-19. Compared to other European countries, the highest percentages of at least one dose uptake of COVID-19 vaccines were observed in Portugal (94.9%), Spain (87.2%), and Iceland (83.3%). The cumulative vaccine uptake in the total population in European countries was 75.6% (data as of 26 January 2023) [15]. The acceptance rate varies over time and may be caused by constantly developing new vaccines, improving the quality and effectiveness of current vaccines, the emergence of different mutations within the SARS-CoV-2 virus, and the spreading of incorrect information from unauthorized parties [23].

Furthermore, due to the fact that SARS-CoV-2 has many similarities to influenza regarding its pathogenicity and respiratory complications [24], respondents were also asked about their willingness to take an influenza vaccination. In addition, this comparison was chosen due to influenza vaccine hesitancy, which is strongly manifested in the general population [25]. Several independent studies reported these concerns increased during the COVID-19 pandemic [26,27]. Most respondents (72.2%) did not take an influenza vaccination. The low influenza vaccination coverage in Poland (61%) was also observed by Zaprutko T et al. [28]. The main concerns are the efficacy, disbeliefs, and misconceptions about the safety and vaccine hesitancy over the years [28].

Since the beginning of the COVID-19 pandemic, influenza epidemiology and surveillance have sharply decreased. The lowest historical level of influenza circulation worldwide was observed in weeks 9 and 10 of 2020 [29]. In Poland, compared to 2019 (before the COVID-19 pandemic), in 2020, 34% fewer influenza-infected patients were registered, while in 2021, this number increased to 37% [30]. This tendency is likely due to social mitigation measures implemented to alleviate the transmission of SARS-CoV-2 infection, which also contribute to the weakening of the transmission of other viral infections, especially those transmitted by similar routes. Another factor contributing to the low influenza circulation is higher influenza vaccination coverage, seen mainly among the age groups at greatest risk of COVID-19 infection. For instance, in Spain, the influenza vaccine uptake increased from an average of 55% in the previous five vaccination campaigns to 64% during the 2020/2021 campaign [31]. In Poland in 2020, only 2.5% more patients were taking the influenza vaccination compared to in 2019. However, in 2021 this number increased to approximately 26% [30]. This result is in line with our data showing the increase in influenza vaccination among those vaccinated against COVID-19 (from 11.3% to 13.3%). Therefore, better coverage in immunization against influenza may positively influence the attitude to COVID-19 vaccination and vice versa. Several studies have shown that the best predictor of the uptake of COVID-19 vaccine is the administration of an influenza vaccine in the previous season [32,33,34]. Furthermore, Conlon et al. determined that patients who took an influenza vaccination during the COVID-19 outbreak (from August 2019 to mid-July 2020) were less likely to be tested as COVID-19 positive. They also found the association between influenza vaccination and decreased COVID-19 mortality and reduced need for intensive care treatment [34]. These and other [24] findings are hence factors leading to an increase in the willingness to take the flu vaccine, which may be potential consequences of alleviating the risk of being COVID-19 infected. 

In our study, 15.7% of all respondents declared anti-vaccine attitudes toward COVID-19. The most frequently cited reasons were fear of post-vaccination complications and their safety (38.0% and 31.8%, respectively, Table 2). This outcome meets the results of other studies concerning the same problem [33,35,36]. In general, a great majority of vaccines have side effects. However, COVID-19 vaccines were approved for use recently; hence, side effects may be different than those found in clinical trials. Consequently, the concerns observed in our study are understandable. It is, therefore, crucial to provide the public with reliable information about the side effects of COVID-19 vaccines [37]. Furthermore, as we know which factors contribute to COVID-19 vaccine refusal, we can propose strategies that should be implemented to increase vaccine acceptance. For instance, Rashid et al. suggested that a few combined interventions, including education, training sessions, and easy vaccine accessibility, may increase influenza vaccine uptake [38]. We believe these strategies may also be useful regarding the COVID-19 vaccine. It is essential for health professionals and medical practitioners to inform patients about the benefits of protecting themselves and their relatives with COVID-19 vaccination.

Studies conducted all over the world highlighted the most critical determinants of intention to take a COVID-19 vaccination, such as age, occupational status, gender, marital status, education level, income, knowledge about COVID-19, past COVID-19 infection, the pre-existence of chronic diseases, as well as physical and mental health conditions [39,40,41,42,43]. In our study, we considered some of the abovementioned sociodemographic factors affecting the attitude toward COVID-19 vaccination. Firstly, we observed positive attitudes toward COVID-19 vaccination among older adults. Respondents over 65 years old were almost four times more likely to accept COVID-19 vaccination than younger adults (OR = 3.69, CI = 95% [3.44–3.96], Table 3). This result is consistent with several other studies reported in the UK, Turkey, Saudi Arabia, Ethiopia, China, and South Africa [40,43,44,45,46,47]. Kilic et al. found a positive relationship between the increase in age and the attitude toward vaccination [44]. Furthermore, in line with our data, the study found a significant relation between education level and positive attitudes toward COVID-19 vaccination [44]. Answers collected in our online questionnaire show that higher education level increased the positive attitudes toward COVID-19 vaccination (OR = 2.14, CI95% [2.07–2.22], Table 3); however, negative attitudes were more frequently observed among respondents with primary and secondary education levels (*p* < 0.001, OR = 2.01, CI = 95% [1.86–2.17], and <0.001, OR = 1.52, CI = 95% [1.41–1.63], respectively, Table 4). In another independent study in Ethiopia, Abebe et al. found the same interplay: age above 46 years or secondary and higher education were significantly associated with COVID-19 vaccine acceptance [48]. In Poland, Raciborski et al. also showed that the lack of higher education is significantly associated with lower willingness to obtain COVID-19 vaccination [20]. Since older people are at the highest risk of severe COVID-19-related complications, they are more afraid to be infected, which in turn increases their willingness to seek vaccination. In addition, highly educated people are more aware of the benefits of prevention in health and have higher receptivity to new health-related information [48]. These results, taken together, show that improving educational status may be one of the general strategies to improve attitudes to vaccinations. Furthermore, advertising and educational campaigns on the safety and efficacy of COVID-19 vaccines should be taken into consideration in order to reach the groups without higher education.

In 2021, Zintel et al. conducted a study comparing 60 reports aiming to determine the role of gender in stating the attitude toward COVID-19 vaccination. A total of 58% of men declared more willingness to take the COVID-19 vaccination compared to their female counterparts [49]. This finding is consistent with several other studies [33,44,50,51], but not with our study. We found that male respondents were more likely to have an anti-vaccine approach compared to females (*p* < 0.001, OR = 2.20, CI = 95% [2.07–2.34], Table 4). However, this study group was not asked about other factors that might affect their final decision, including net income or occupation. There were also no questions about addictions and smoking history. Furthermore, it sounds reasonable that more male respondents are against COVID-19 vaccination due to their “laid-back” approaches to COVID-19 vaccination, which in turn, decreases their awareness about the health crisis caused by COVID-19. This phenomenon was observed more often among male respondents declaring very good physical condition (OR = 1.51 CI95% [1.15–1.99]). Nevertheless, additional research is needed on gender regarding COVID-19 vaccine hesitancy.

This study has several limitations. First, this study was based on the results of an online survey. Therefore, we are forced to believe in the sincerity of the participants filling in the questionnaire. It is also very difficult to determine the percentage of uncompleted questionnaires at each stage of the research. Secondly, the study was conducted in a period of almost two years. Public opinion may change because of media campaigns and vaccination promotions by public authorities and medical professionals. As the survey was anonymous, it was not possible to inform participants of the results of the study or provide psychological support if necessary. The study group is not representative of Polish society despite the fact that the questionnaire was distributed to various general groups. In order to reduce this risk, the online questionnaire was spread around social media for different groups of interest. 

The lack of knowledge regarding potential vaccine complications and their safety should constitute essential targets for educational programs in the Polish population. The aim is to alleviate the COVID-19 pandemic crisis and enhance vaccination rates [52]. The healthcare system plays a primary role in this task: the global challenge is to educate, inform, and intervene to increase positive attitudes toward COVID-19 vaccination. The results of this study may motivate public benefit organizations and local authorities in Poland to reach specific groups, provide reliable knowledge about the importance of COVID-19 vaccinations, and reduce COVID-19 vaccine hesitancy.

## Figures and Tables

**Figure 1 vaccines-11-00700-f001:**
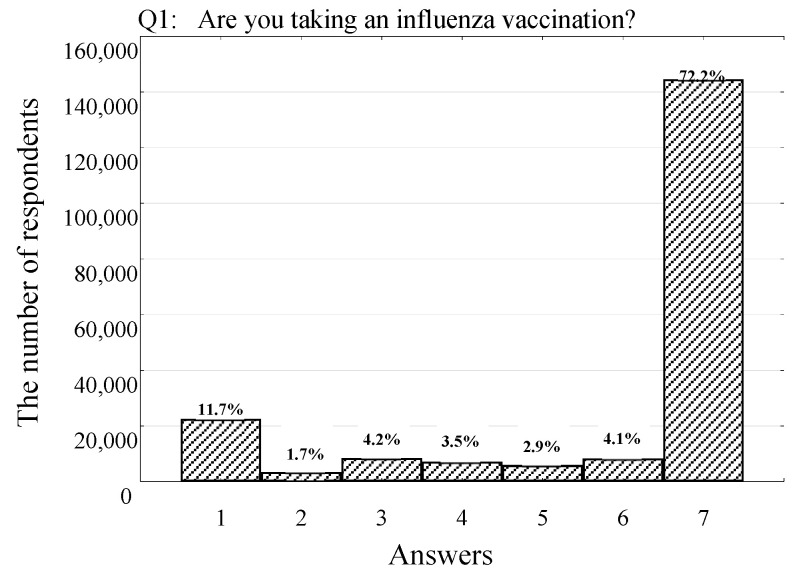
Number (percentage) of respondents declaring their attitude toward influenza vaccination among 200,000 Poles participating in the survey in 2022 (each point represents the answer shown in Table 2: 1—yes, every year, including in 2022; 2—yes, every year, but I couldn’t take an influenza vaccination in 2022 due to the lack of its availability; 3—yes, but not every year. I got the influenza vaccine in 2022; 4—yes, but not every year. I wanted to get a vaccine in 2022, but I couldn’t due to the lack of its availability; 5—usually not, but I got the influenza vaccine in 2022; 6—usually not. I wanted to get a vaccine in 2022, but I couldn’t due to the lack of its availability; 7—no, never).

**Figure 2 vaccines-11-00700-f002:**
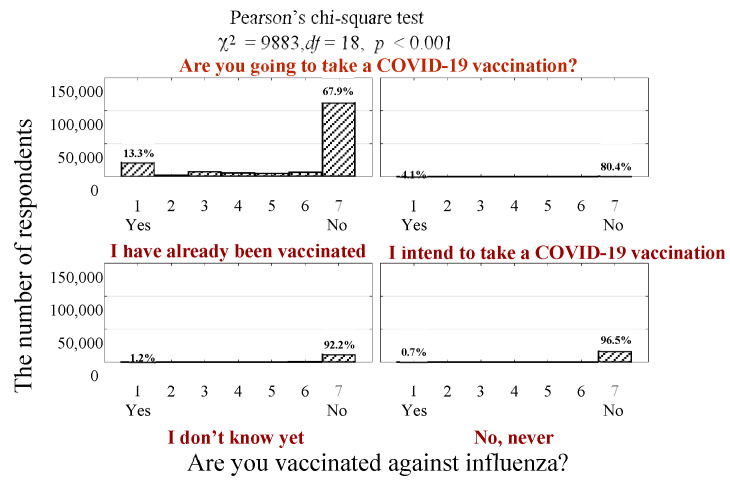
Number (percentage) of respondents answering differently to questions about COVID-19 and influenza vaccination.

**Figure 3 vaccines-11-00700-f003:**
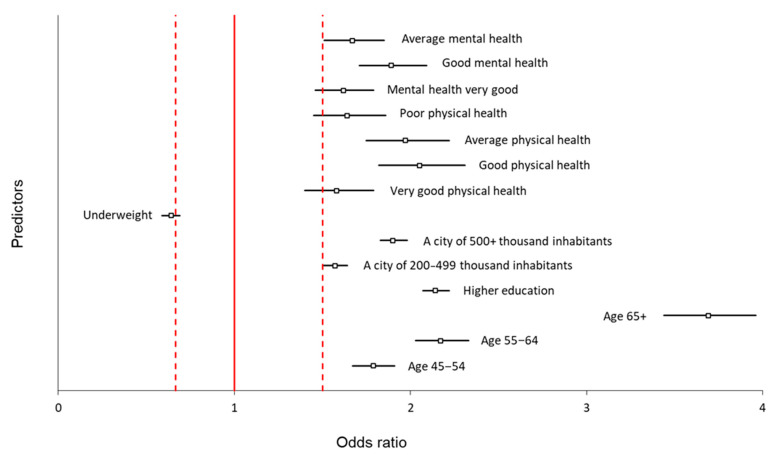
Odds ratios and their 95% confidence intervals for significant predictors of positive attitude towards COVID-19 vaccination. Rectangles show the odds ratio of each parameter, the horizontal black lines show their 95% confidence intervals, and the red vertical lines show odds ratio ranges lower and higher than 1.5 times compared to the reference values.

**Table 1 vaccines-11-00700-t001:** General characteristics of the subjects.

Feature (Variable)	*n*	Percentage
Gender		
Men	119,169	59.6%
Women	80,831	40.4%
Age (years old)		
18–24	5011	2.5%
25–34	19,005	9.5%
35–44	36,333	18.2%
45–54	43,265	21.6%
55–64	43,624	21.8%
65+	52,765	26.4%
Education		
Primary	27,154	13.6%
Secondary	59,384	29.7%
Higher	112,462	56.2%
Place of residence		
Village	44,232	22.1%
Town less than 19,000 inhabitants	23,824	11.9%
Town between 20,000 to 49,000 inhabitants	30,729	15.6%
Town between 50,000 to 99,000 inhabitants	23,293	11.6%
Town between 100,000 to 199,000 inhabitants	22,047	11.0%
Town between 200,000 to 499,000 inhabitants	21,203	10.6%
Town more than 500,000 inhabitants	34,672	17.4%
Region in Poland		
South	45,452	22.7%
Northwest	32,348	16.2%
Southwest	22,008	11.0%
North	28,000	14.0%
Central	17,817	8.9%
East	22,866	11.4%
Masovian voivodeship	31,508	15.7%
BMI		
Underweight	3408	1.7%
In norm	68,471	34.2%
Overweight	77,630	38.8%
Obesity	50,489	25.2%
Physical health		
Very good	27,606	3.8%
Good	80,417	40.2%
Normal	76,259	38.2%
Bad	14,285	7.1%
Very bad	1433	0.7%
Mental health		
Very good	45,221	22.6%
Good	83,660	41.8%
Normal	52,354	26.2%
Bad	16,642	8.3%
Very bad	2123	1.1%

**Table 2 vaccines-11-00700-t002:** Number (percentage) of respondents in groups differing in their assessment of approach to vaccination among 200,000 participating in the study.

**Q1: Are you vaccinated against influenza?**	** *n* **	**(%)**
1—Yes, every year, including in 2022	22,512	11.3
2—Yes, every year, but I couldn’t take an influenza vaccination in 2022 due to the lack of its availability	3413	1.7
3—Yes, but not every year. I got the influenza vaccine in 2022	8431	4.2
4—Yes, but not every year. I wanted to get a vaccine in 2022, but I couldn’t due to the lack of its availability	7069	3.5
5—Usually not, but I got the influenza vaccine in 2022	5873	2.9
6—Usually not. I wanted to get a vaccine in 2022, but I couldn’t due to the lack of its availability	8233	4.1
7—No, never	144,469	72.2
**Q2: Are you going to take a vaccine against COVID-19?**	** *n* **	**(%)**
1—Yes, I have already been vaccinated	166,445	83.2
2—Yes, I intend to take a COVID-19 vaccination	2217	1.1
3—I don’t know yet	13,471	6.7
4—No, never	17,867	8.9
**Q3: Why won’t you take a COVID-19 vaccination?**	** *n* **	**(%)**
1—I have concerns about the safety of the COVID-19 vaccine	9966	5.0
2—I am afraid of post-vaccination complications	11,913	6.0
3—I can’t get vaccinated due to medical reasons	4372	2.2
4—I am against vaccination in general	5087	2.5

**Table 3 vaccines-11-00700-t003:** Evaluation of positive approaches toward COVID-19 vaccination among studied respondents characterizing different sociodemographic factors (odds ratios higher than 2 were bolded).

Predictors of Positive Attitudes toward COVID-19 Vaccination	Attitude to Vaccination				*p*	OR (95%)
	Positive		Negative			
	*n*	(%)	*n*	(%)		
**Gender**:						
Men	69,364	41.1%	11,467	36.6%	<0.001	1.21 (1.18–1.24)
Women	99,298	58.9%	19,871	63.4%		1.00 (ref.)
**Age (years old):**					<0.001	
18–24	3727	2.2%	1284	4.1%		1.00 (ref.)
25–34	14,243	8.4%	4762	15.2%		1.03 (0.96–1.11)
35–44	28,504	16.9%	7829	25.0%		1.25 (1.17–1.34)
45–54	36,275	21.5%	6990	22.3%		1.79 (1.67–1.91)
55–64	37,650	22.3%	5971	19.1%		**2.17 (2.03–2.33)**
65+	48,263	28.6%	4502	14.4%		**3.69 (3.44–3.96)**
**Education**:					<0.001	
Primary	20,736	12.3%	6418	20.5%		1.00 (ref.)
Secondary	48,778	28.9%	10,606	33.8%		1.42 (1.37–1.47)
Higher	99,148	58.8%	14,314	45.7%		**2.14 (2.07–2.22)**
**Place of residence:**					<0.001	
Village	35,382	21.0%	8850	28.2%		1.00 (ref.)
Town less than 19,000 inhabitants	20,132	11.9%	3692	11.8%		1.36 (1.31–1.42)
Town between 20,000 to 49,000 inhabitants	25,790	15.3%	4939	15.8%		1.31 (1.26–1.36)
Town between 50,000 to 99,000 inhabitants	19,606	11.6%	3687	11.8%		1.33 (1.28–1.39)
Town between 100,000 to 199,000 inhabitants	18,825	11.2%	3222	10.3%		1.46 (1.40–1.53)
Town between 200,000 to 499,000 inhabitants	18,288	10.8%	2915	9.3%		1.57 (1.50–1.64)
Town more than 500,000 inhabitants	30,639	18.2%	4033	12.9%		1.90 (1.83–1.98)
**Region in Poland:**					<0.001	
South	37,854	22.4%	7598	24.2%		1.00 (ref.)
Northwest	27,852	16.5%	4494	14.3%		1.24 (1.20–1.29)
Southwest	18,463	10.9%	3545	11.3%		1.05 (1.00–1.09)
North	24,199	14.3%	3801	12.1%		1.28 (1.23–1.33)
Central	14,974	8.9%	2843	9.1%		1.06 (1.01–1.11)
East	17,867	10.6%	5002	16.0%		0.72 (0.69–0.75)
Masovian voivodeship	27,453	16.3%	4055	12.9%		1.36 (1.30–1.42)
**BMI**					<0.001	
Underweight	2553	1.5%	855	2.7%		0.64 (0.59–0.69)
In norm	564,56	33.5%	12,015	38.3%		1.00 (ref.)
Overweight	66,544	39.5%	11,086	35.4%		1.28 (1.24–1.31)
Obesity	43,109	25.6%	7382	23.6%		1.24 (1.20–1.28)
**Physical health**					<0.001	
Very good	22,568	13.4%	5038	16.1%		1.58 (1.40–1.79)
Good	68,601	40.7%	11,816	37.7%		**2.05 (1.82–2.31)**
Normal	64,680	38.3%	11,579	36.9%		1.97 (1.75–2.22)
Bad	11,754	7.0%	2531	8.1%		1.64 (1.45–1.86)
Very bad	1059	0.6%	374	1.2%		1.00 (ref.)
**Mental health**					<0.001	
Very good	37,790	22.4%	7431	23.7%		1.62 (1.46–1.79)
Good	71,630	42.5%	12,030	38.4%		1.89 (1.71–2.09)
Normal	44,002	26.1%	8352	26.7%		1.67 (1.51–1.85)
Bad	13,629	8.1%	3013	9.6%		1.44 (1.29–1.60)
Very bad	1611	1.0%	512	1.6%		1.00 (ref.)

The bolding in OR indicates a significant statistical change higher than 2.00 (info above the table).

**Table 4 vaccines-11-00700-t004:** Evaluation of negative approaches toward COVID-19 vaccination among studied respondents characterizing different sociodemographic factors (odd ratios higher than 2 were bolded).

Predictors of Negative Attitude to COVID-19 Vaccination	Respondents against Vaccination				*p*	OR (95%)
	Yes		No			
	*n*	(%)	*n*	(%)		
**Gender**:						
Men	2673	52.5%	8794	33.5%	<0.001	**2.20 (2.07–2.34)**
Women	2414	47.5%	17,457	66.5%		1.00 (ref.)
**Age (years old):**					<0.001	
18–24	269	5.3%	1015	3.9%		1.36 (1.16–1.59)
25–34	832	16.4%	3930	15.0%		1.09 (0.97–1.21)
35–44	1306	25.7%	6523	24.8%		1.03 (0.93–1.13)
45–54	1058	20.8%	5932	22.6%		0.92 (0.83–1.01)
55–64	888	17.5%	5083	19.4%		0.90 (0.81–1.00)
65+	734	14.4%	3768	14.4%		1.00 (ref.)
**Education level:**					<0.001	
Primary	1427	28.1%	4991	19.0%		**2.01 (1.86–2.17)**
Secondary	1879	36.9%	8727	33.2%		1.52 (1.41–1.63)
Higher	1781	35.0%	12,533	47.7%		1.00 (ref.)
**Place of residence:**					0.061	
Village	1446	28.4%	7404	28.2%		1.00 (ref.)
Town less than 19,000 inhabitants	607	11.9%	3085	11.8%		1.01 (0.91–1.12)
Town between 20,000 to 49,000 inhabitants	763	15.0%	4176	15.9%		0.94 (0.85–1.03)
Town between 50,000 to 99,000 inhabitants	653	12.8%	3034	11.6%		1.10 (1.00–1.22)
Town between 100,000 to 199,000 inhabitants	524	10.3%	2698	10.3%		0.99 (0.89–1.11)
Town between 200,000 to 499,000 inhabitants	435	8.6%	2480	9.4%		0.90 (0.80–1.01)
Town more than 500,000 inhabitants	659	13.0%	3374	12.9%		1.00 (0.90–1.11)
**Region in Poland:**					0.213	
South	1225	24.1%	6373	24.3%		1.00 (ref.)
Northwest	711	14.0%	3783	14.4%		0.98 (0.88–1.08)
Southwest	612	12.0%	2933	11.2%		1.09 (0.98–1.21)
North	647	12.7%	3154	12.0%		1.07 (0.96–1.18)
Central	440	8.6%	2403	9.2%		0.95 (0.85–1.07)
East	825	16.2%	4177	15.9%		1.03 (0.93–1.13)
Masovian voivodeship	627	12.3%	3428	13.1%		0.95 (0.86–1.06)
**BMI**					0.196	
Underweight	145	2.9%	710	2.7%		1.09 (0.90–1.31)
In norm	1900	37.4%	10,115	38.5%		1.00 (ref.)
Overweight	1861	36.6%	9225	35.1%		1.07 (1.00–1.15)
Obesity	1181	23.2%	6201	23.6%		1.01 (0.94–1.10)
**Physical health**					<0.001	
Very good	1215	23.9%	3823	14.6%		1.51 (1.15–1.99)
Good	1987	39.1%	9829	37.4%		0.96 (0.73–1.26)
Normal	1517	29.8%	10,062	38.3%		0.72 (0.55–0.94)
Bad	303	6.0%	2228	8.5%		0.65 (0.48–0.87)
Very bad	65	1.3%	309	1.2%		1.00 (ref.)
**Mental health**					<0.001	
Very good	1583	31.1%	5848	22.3%		1.34 (1.06–1.70)
Good	1878	36.9%	10,152	38.7%		0.92 (0.72–1.16)
Normal	1145	22.5%	7207	27.5%		0.79 (0.62–1.00)
Bad	395	7.8%	2618	10.0%		0.75 (0.58–0.96)
Very bad	86	1.7%	426	1.6%		1.00 (ref.)

The bolding in OR indicates a significant statistical change higher than 2.00 (info above the table).

## Data Availability

Data supporting reported results can be found at https://narodowytestzdrowia.medonet.pl/ (accessed on 26 January 2023).

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
