# Peer review of "Evaluation of the Approach towards Vaccination against COVID-19 among the Polish Population—In Relation to Sociodemographic Factors and Physical and Mental Health"

_vaccines, 2023, doi:10.3390/vaccines11030700_

Round 1

Reviewer 1 Report

The manuscript entitled "Evaluation of the approach towards vaccination against  COVID-19 among the Polish population - correlation with sociodemographic factors and physical and mental health" by GoÅ‚Ä™biowska and colleagues aims to determine socio-demographic factors contributing to the attitude to vaccination against COVID-19. 

This is a very important topic as the COVID-19 pandemic demonstrated the importance of vaccines and vaccinations, and a correct information about these topics. 

I think that some issues need to be addressed before re-considering the manuscript: 

- the manuscript needs extensive editing of english language and style (please avoid repetitions and make sure the numbers are correctly written, e.g. with the correct separators for decimals or thousands)

- Parts of the introduction seem untied (e.g. "The total number of confirmed cases in Poland reached more than 6 million, with over 120 000 deaths [1]. Therefore the COVID-19 disease was declared by the World Health Organization (WHO) in march 2020 as a global pandemic [2]". It seems you are suggesting that the WHO declared COVID-19 a pandemic BECAUSE of the high number of cases in Poland, which is not true). Please review the whole manuscript in this sense. 

- please provide the entire text of the survey (socio-demographic data requested, questions etc), as an additional file, for example as supplementary material (possibly translated in English, it does not matter if it's not validated in english as you administrated it in another language). 

- I am not sure that Table 1 should be placed in the methods section, as it shows results. 

Author Response

Thank you for providing insightful feedback on ways to strengthen our paper. It is with great pleasure that we resubmit our article for further consideration. We have incorporated changes that reflect the detailed suggestions you have graciously provided. We also hope that our edits and the responses we provided below satisfactorily address all the issues and concerns you have noted.

Reviewer 2 Report

Authors performed a study to explore the factors that affect people’s decision to accept COVID-19 vaccination in Poland. Unfortunately, the study suffers from serious flaws. The manuscript suffers from a great number of disadvantages that decrease the validity of the results. Then, you can find a summary of these disadvantages.

Please, use the term “relation” instead of correlation in the title and other parts of the manuscript.

In general, the manuscript would benefit from careful editing throughout to ensure full sentences and proper grammar are used. A native English speaker should read the manuscript and correct it. There are many misunderstandings. Many parts of the manuscript are totally confusing. For example the term “mortalities” in line 37, the term “march” in line 39, the term “correlated” in line 44, etc.

Introduction

The Introduction section is very poor. You write “Nevertheless, little is known about the rationale and factors influencing the attitude 58 toward COVID-19 vaccination” but it is not correct. There is a tremendous amount of literature on this field. Please, describe and present this literature in a few paragraphs. Please, focus on the factors that influence individuals’ decision to accept COVID-19 vaccination. Also, you should show in the Introduction the gap in the literature that your study will fill. What does your study add in the literature?

Methods

How can you prove that your sample is representative?

Please, use the correct terms in your methodology. Since the questionnaire was filled online, it was impossible to conduct interviews (see line 76).

Since the questionnaire was filled online, how did you obtain the verbal consent of the participants (line 78)?

Since your questionnaire (reference 8) is in Polish, international readers cannot have access to this. Could you please provide an English version of the questionnaire as a supplement?

How did you apply the stratification sampling?

Tables 1 and 2 should be removed on Results section instead of Methods section. Also lines 73 and 74 should be removed on the Results section.

How did you measure people that have already been vaccinated against seasonal influenza?

Why did you consider as significant predictors of negative or positive attitudes towards COVID-19 vaccination were those whose odds ratios were outside the range of 1.5 times the reference values? Does the literature support this? Could you please provide some references? Please, explain the reason that you did not use the conventional level a = 0.05.

Please, explain the reason that you did not use valid instruments to measure physical and mental health such as SF-12?

Please, perform multivariable analysis in order to eliminate confounding.

Results

Please, do not repeat in detail the results of tables in text. For example, just present the percentage of males/females instead of numbers.

Line 132. Body mass index is not an index of malnutrition.

Please remove figures 1, 2 and 3 and present the information in the text using tables.

Please, perform multivariable analysis and present the adjusted odds ratios.

Discussion

After you perform multivariable analysis, please discuss the independent variables that affect individuals’ attitudes. Thus, lines 271-310 should be probably rewritten.

Lines 244-236. COVID-19 definitely does not have similar morbidity and mortality rates with influenza.

Please, do not repeat results on the Discussion section (e.g. line 257).  

Please add the limitations of your study.

Please present the public health implications of your study in Poland and worldwide.

Author Response

(The authors gave the same response as above.)

Round 2

Reviewer 1 Report

Dear Authors, many thanks for revising this manuscript. Readability and overall quality improved, but some issues still need to be addressed: 

- please revise all numbers in your manuscript and make sure that "," is always used to separate thousands, and "." is used for decimals (e.g. you need to check and correct all the tables. 

- The % of people vaccinated against influenza you found in your study is consistent with national data reported by another study [1]. I suggest expanding this topic, also reporting the literature speculating about a possible association between flu vaccine acceptance and COVID-19 vaccine acceptance and the reasons for it [e.g. 2-3]

- You speculate about "greater knowledge and awareness about positive roles of vaccination" in those immunized against COVID-19 (lines 284-285). Despite agreeing with this view, my feeling is that this is coming a bit out of the blue. I suggest expanding the concept of vaccine literacy [e.g. 4], which is complex and has many implications. 

- I tend to disagree with the statement that your findings regarding gender-related COVID-19 vaccine acceptance are contrary to most studies (lines 332-333); many studies report female subjects as more inclined to accept COVID-19 vaccines. As you cite primary studies, I suggest carefully reviewing the evidence body looking for systematic reviews on this topic to confirm or change what is stated. 

1. https://pubmed.ncbi.nlm.nih.gov/34915972/

2. https://pubmed.ncbi.nlm.nih.gov/35148322/

3. https://pubmed.ncbi.nlm.nih.gov/35207628/

4. https://pubmed.ncbi.nlm.nih.gov/36794338/

Author Response

(The authors gave the same response as above.)

Reviewer 2 Report

Dear Authors thank you for the changes you make in your manuscript.

Author Response

Thank you for providing insightful feedback on ways to strengthen our paper.

Round 3

Reviewer 1 Report

Thank you for addressing all comments.